# The Effect of Industrial Structure Upgrading and Human Capital Structure Upgrading on Green Development Efficiency—Based on China's Resource-Based Cities

Wanfang Shen [1,2], Yufei Liu [3], Xiaowen Liu [4], Jianing Shi [3], Wenbin Liu [5] and Chengye Liu [6,7,*]

1   Shandong Key Laboratory of Blockchain Finance, Shandong University of Finance and Economics, Jinan 250014, China
2   Joint Research Center for Decision Making and Evaluation, Shandong University of Finance and Economics, Jinan 250002, China
3   School of Mathematics and Quantitative Economics, Shandong University of Finance and Economics, Jinan 250002, China
4   Credit Card Center of China Minsheng Banking Corporation Limited, Beijing 100071, China
5   Division of Business and Management, Beijing Normal University-Hong Kong Baptist University United International College, Zhuhai 519087, China
6   Institute of Automation, Qilu University of Technology (Shandong Academy of Sciences), Jinan 250014, China
7   Key Laboratory of UWB & THz of Shandong Academy of Sciences, Shandong Academy of Sciences, Jinan 250014, China
*   Correspondence: liucy@sdas.org

## Highlights:

**What are the main findings?**

- GDE in China's resource-based cities was relatively low during 2010-2019.
- ISU, HCSU and their interaction can improve GDE in China's resource-based cities.

**What is the implication of the main finding?**

- There exists heterogeneity in the direct effect and interaction effect of ISU and HCSU on four types of resource-based cities.
- Government should pay more attention to the green development of China's resource-based cities.

**Abstract:** Green development is crucial to global natural resource conservation, environmental improvement and sustainable development. Furthermore, resource-based cities' green development is more challenging compared with that of other types of cities. On such basis, it is a necessity to understand the green development level of such cities. Therefore, we introduce green development efficiency (GDE), which is a key indicator for measuring green development. This paper takes China's 112 resource-based cities during 2010–2019 as its research object, and examines their GDE using the Super-SBM-Undesirable model. Moreover, industrial structure upgrading (ISU) and human capital structure upgrading (HCSU) have important implications for green development. To further explore the influence of ISU and HCSU on GDE, this paper employs a fixed effect model, an interaction effect model and a threshold model. Finally, considering the differences between different resource-based cities, the heterogeneity of ISU and HSCU on GDE in four types of China's resource-based cities is also explored. It is found that (1) although GDE is on the track of steady improvement, the overall GDE was still relatively low during 2010–2019, with an average GDE of about 0.8; (2) ISU, HCSU and their interaction can promote GDE in resource-based cities and with the intensity of industrial structure increasing, the interaction effect of ISU and HSCU on GDE in resource-based cities shifts from positive to negative; (3) there exists heterogeneity in the direct effect and interaction effect of ISU and HCSU among four types of resource-based cities (i.e., mature cities, growing cities, declining cities and regenerating cities). Our findings offer a data reference for the green and sustainable development of China's resource-based cities, and also a method reference for other countries' resource-based cities.

**Keywords:** green development efficiency (GDE); Super-SBM-Undesirable model; human capital structure upgrading (HCSU); industrial structure upgrading (ISU); resource-based cities

## 1. Introduction

In 2015, global sustainable development goals (SDGs) were proposed by the United Nations, aiming to "free humanity from poverty, secure a healthy planet for future generations, and build peaceful, inclusive societies as a foundation for ensuring lives of dignity for all" [1]. Among them, a major challenge is sustainable economic development [2]. Economic growth is significant for improving living standards and beneficial in helping free humanity from poverty. However, it is also a dominant cause of environment deterioration. Taking China as an example, since the reform and opening up, China's social and economic development has attained remarkable achievements. According to China's National Bureau of Statistics, China's GDP grew at an average annual rate of 6.6% from 2013 to 2021. However, the traditional development mode takes economic interests as guidance, neglecting the protection of resources and the environment. As a result, it brings about environmental pollution, which goes against SDGs. Today, the situation of China's ecological environment is that the ecological deficit is gradually expanding to a certain extent. Conflicts between economic development and the preservation of the natural environment appear and heavily restrict socioeconomic sustainable development. To achieve SDGs and resolve these conflicts, it is urgent for China to attach importance to promoting a green economy and accelerating the pace of sustainable development.

As an important part of sustainable development, green development can effectively resolve conflicts between economic development and the natural environment. Specifically, green development is defined as a mode of economic growth and social development which is aimed at efficiency, harmony and sustainability [3]. Green development efficiency (GDE) as a manifestation of the coordinated and sustainable development of humanity and nature [3] has played an increasingly prominent role in green development [4]. The research objects on green development in the existing literature range widely, including country-level [5], province-level [6–8] and city-level objects [9,10]. However, there exists a particular type of city which plays an important role in providing material resources for economic development: the resource-based city. Specifically, the resource-based city refers to a kind of city that relies on the exploitation and processing of abundant natural resources (forest resources, mineral resources, etc.) in the early stage of urban development, and has formed a single and rugged development mode [11]. These cities' sustainable development is more challenging and should be paid more attention compared with that of other types of cities [12]. Natural resource curse theory and its transmission mechanisms [13,14], such as the Dutch disease [15], the institutional weakening effect [16] and the crowding-out effect [17], have been used to explain these cities' problems of unsustainable development [18]. In addition, in China, there are 126 resource-based cities, accounting for 43% prefecture-level cities in this country [19]. This indicates that achieving sustainable and green development in resource-based cities is more urgent and challenging in China compared with other countries. However, to our best knowledge, current studies on such cities mostly focus on theoretical analysis [20,21], or conducting green development evaluation for only one type of resource-based cities (e.g., oil-resource-based cities, gas-resource-based cities) [22,23]. Nonetheless, a quantitative evaluation of all resource-based cities has not been explored. To fill this gap, this paper examines GDE in resource-based cities in China using the DEA model. We hope that our conclusions can help the government issue more effective policies to guide the green development of such cities.

Meanwhile, there are some scholars further exploring the influencing factors of GDE to improve green development. For example, ref. [7] measured the industrial green technology innovation efficiencies of 29 provinces in China using a two-stage network SBM-Undesirable-DEA model and explored the effect of environmental regulation on this efficiency, proving "Porter's hypothesis" is established in China. Ref. [24] tested the impact of digital economy on GDE and found that it significantly promotes GDE in China. Ref. [25] explored the effect of industrial structure adjustment on GDE and proposed that the up-

grading of industrial structure has a greater effect on GDE compared with industrial structure rationalization.

Among the influencing factors, industrial structure upgrading (ISU) is regarded as an effective way to promote and achieve green development according to the 13th five-year plan of national economic and social development in China (2016–2020). Industrial structure is referred to as a composition of industries. Industrial structure upgrading (ISU) is defined as the process of industrial structure changing from low form to high form, which is one of the essential tools of green economic growth [26]. In recent years, China's industrial structure has changed significantly. According to the National Bureau of Statistics of China, from 2013 to 2021, the average annual value-added growth rate of the tertiary industry reached 7.4%, 0.8% higher than the average annual growth rate of the GDP. Its annual contribution to economic growth reached 55.6%. However, relevant studies that explore the influence of ISU on economic growth or environmental protection are limited. Few studies combine ISU and green development. To fill this gap, this paper further explores the impact of ISU on GDE.

Furthermore, human capital structure upgrading (HCSU), characterized by the dynamic evolution from primary human capital to senior human capital, has also played an important role in economic growth [26]. The mediating effect or moderating effect transmission path of "HCSU → ISU → sustainable economic development" and the influencing path of "ISU → HCSU" have been confirmed by scholars [6,27]. However, there is a lack of research on the interaction effect of ISU and HCSU on GDE. To fill this gap, this paper further explores the interaction effect of these two factors on GDE in addition to the direct effect. In addition, exploring the relationship between ISU, HCSU and GDE in resource-based cities is of practical significance for regulating the development path of such cities.

Finally, considering that the foundations and driving forces of four types of resource-based cities in China (i.e., mature cities, growing cities, declining cities and regenerating cities) are different, we further conduct a heterogeneity analysis based on the types of such cities, which is ignored in the previous studies on resource-based cities. It is more conducive to the green development of such cities to put forward development suggestions for each type of cities accordingly.

In summary, this paper makes three main contributions. Firstly, we calculate the GDE in resourced-based cities during 2010–2019 using the Super-SBM-Undesirable model and further analyze its dynamic evolution. Secondly, the intermediary effect and threshold effect of ISU and HCSU on the GDE in resource-based cities are explored. Thirdly, we conduct a heterogeneity analysis based on the type of resource-based cities and propose specific suggestions for each type of cities for their own green development. The purpose of this paper is to expand the research on GDE to resource-based cites in China and help the government and relevant enterprises implement more valid policies for such cities in different stages of green development.

The remainder of this paper is organized as follows. In Section 2, we propose our research hypotheses based on theoretical analysis. Section 3 builds models, including an efficiency measurement model and econometric models. Section 4 reports the empirical results. In Section 5, we conclude and offer corresponding policy suggestions. Lastly, we discuss the study's limitations and propose future research in Section 6.

## 2. Literature Review

Before calculating GDE and exploring relevant influencing factors, we recognize the necessity to discuss the existing literature to elaborate what has been achieved in terms of GDE.

Green development was first proposed by the United Nations Development Programme (UNDP) in 2002. Soon after, scholars began to study the process of green progress and prospects in different countries, such as Malaysia [28], Oman [29] and Bangladesh [30]. There is also a study from the global perspective [31]. The above studies mostly focused on

theoretical analysis. To quantify green development, GDE was introduced and explored, especially in China. The main calculation method was the DEA model [32]. To our best knowledge, GDE was firstly calculated by [33]. They applied the Super-efficiency-CCR model to calculate GDE in 31 regions in China based on the equation of "GDE = green output/ green input". However, this model cannot incorporate pollution and other undesirable outputs. To overcome this difficulty, [25] introduced the Super-efficiency SBM model with undesirable outputs and concluded that China's GDE had upward, downward and U-shaped trends from 1999 to 2017, taking provincial data from China as the object. After this, the SBM model with undesirable outputs became the main model for calculating GDE and scholars began to study GDE based on this model through different research objects. Ref. [34] constructed urban GDE in China from 2005 to 2015 and found that its evolution had the characteristics of a "W-shaped" pattern. Cui and Liu estimated the urban GDE of 12 cities in the Jing-Jin-Ji region of China in 2013–2017 and concluded that GDE declined steadily in this period [35]. Ref. [36] estimated the GDE of 34 cities in Northeast China and found it exhibited a fluctuant upward trend in 2013–2016. Refs. [37,38] examined GDE based on the data of 61 prefecture-level cities in the Yellow River Basin in China from 2005 to 2017, while [39] was interested in the Yangtze River Economic Belt in China. There are also some studies focusing on specific industries, such as the tourism industry [40], the chemical industry [41] and the cultural industry [42].

Overall, GDE has been extensively studied using the DEA model with a wide range of research objects. However, to our best knowledge, these were mainly from the macro scale of the whole country, province or specific region, but studies on GDE in resource-based cities are scarce. As is mentioned in the previous section, such cities' sustainable development is more challenging, so their green development should be paid more attention. Therefore, this paper estimates GDE in these cities to understand their green development level.

GDE has been extensively discussed, as is shown in the previous subsection. Subsequently, different econometric models have been applied to further explore the influencing factors of GDE. The main influencing factors include the economic development level, ISU, HCSU, the degree of openness, technological innovation, pollution control investment and foreign direct investment.

To our best knowledge, the economic development level has been verified to be positively related to GDE [34–36]. The degree of openness, pollution control investment and foreign direct investment can also promote GDE [35,36], while human capital has a negative effect on GDE [25]. Technological innovation has no significant influence [35]. However, the influence of industrial structure on GDE is controversial. Ref. [35] concluded that industrial structure has a positive influence while [36] found its influence to be negative.

There are also studies on the impact of the interaction between ISU and HCSU. To our best knowledge, ref. [43] firstly recognized the interaction between aggregate changes in human capital and structural change. Then, scholars began to explore the interaction's influence on different aspects, especially economic growth. Ref. [44] explored the effect of the interaction of human capital and structural changes on economic growth during the period of 1986–2015 for the MENA region countries and concluded that there was no significant influence. Ref. [45] concluded that over a longer time span and for more highly developed (OECD) countries, the impact of the interaction between the two on economic growth was positive. Furthermore, the mediating effect or moderating effect transmission path of "HCSU → ISU → sustainable economic development" and the influencing path of "ISU → HCSU" have been confirmed by scholars [6,27,46,47]. However, studies on the influence of the interaction between ISU and HCSU on green development are scarce. To fill this gap, we further explore this relationship.

On such basis, in this paper, we are interested in how ISU and HCSU influence GDE and whether there is an interaction effect between the two on GDE. In addition, considering the characteristics of our research object, we also explore whether heterogeneity exists in the impacts of ISU, HCSU and their interaction on the GDE of four types of resource-based cities.

## 3. Theoretical Analysis and Research Hypotheses

**Hypothesis 1.** *HCSU can improve the GDE.*

The structuralist school believes that structural changes in technology are crucial to economic growth [48]. In addition, an optimal technological structure is endogenously determined by its factor endowment structure, such as human capital [48,49]. Moreover, Lucas believes that specialized human capital can not only produce increasing marginal output, but also overcome the diminishing marginal output effect of other production factors through externalities. This can lead to the promotion of overall productivity [50,51].

In addition, the strong environmental awareness of high-quality human capital can increase citizens' participation in ecological governance behavior, producing a resource-saving effect and reducing environmental pollution.

Combined with the above analysis, HCSU can ultimately promote GDE.

**Hypothesis 2.** *ISU can improve the GDE.*

Firstly, in the process of ISU, resource allocation can be optimized. In other words, resources are transferred from sectors with low productivity to those with high productivity or large profits. This generates "structural dividends", promoting economic growth [52].

Secondly, technological progress is the primary driver and realization path of ISU and economic growth [50,53]. Green high-tech enterprises can avoid fierce competition in existing markets, grasp new opportunities and achieve high-quality growth [54]. This process can weed out the enterprises with low technological development and high pollution or promote these enterprises' transition. This will eventually lead to ISU. In return, ISU can further boost the spillover of existing green technologies through industrial interaction and integration, reducing the dependence on natural resources such as energy and improving the efficiency of resource utilization. Therefore, ISU can ultimately improve the overall GDE [50].

**Hypothesis 3-1.** *The interaction effect between ISU and HCSU exists. When the two match each other, the GDE will be improved; otherwise, the GDE will be reduced.*

**Hypothesis 3-2.** *ISU and HCSU have a threshold effect on their interaction effect on GDE.*

Firstly, HCSU can promote ISU by driving technological progress and consumer demand. On the one hand, as mentioned above, technological innovation, which is accompanied by high-level human capital, is an essential driving force for ISU. On the other hand, advanced human capital usually has relatively higher income levels and consumer preferences for tertiary and high-tech industries. According to the principle of supply and demand, this attracts related sectors, thus promoting ISU.

Secondly, ISU can promote HCSU through the demand effect and educational training. With the upgrading and transformation of industrial structure, industries have an increasing demand for matched high-quality human capital. To meet the needs of social and national development, social wealth is increasingly concentrated in education, thus promoting HCSU.

Lastly, when ISU and HCSU are effectively matched, it is helpful for them to play a greater role in promoting GDE, producing positive interaction. However, when either of them develops too slowly, the other's development needs cannot be met, resulting in a hollowing-out state, such as brain drain, unbalanced employment structure or untenable ISU. This produces a negative interaction. Moreover, the intensity of the two may affect the matching degree, and then generate a nonlinear effect of interaction on GDE.

**Hypothesis 4.** *Heterogeneity exists in the impacts of ISU, HCSU and their interaction on the GDE of mature, growing, declining and regenerating cities.*

According to the National Plan for Sustainable Development of Resource-based Cities (2013–2020) (from now on referred to as the Plan) [11], there are four types of resource-based cities in China: mature, growing, declining and regenerating cities. The driving forces and foundations of green development in the four types of cities are different. On such basis, we propose the hypothesis that heterogeneity exists in the impacts of ISU, HCSU and their interaction on the GDE of mature, growing, declining and regenerating cities.

## 4. Materials and Methods

### 4.1. Data Sources and Variables

#### 4.1.1. Data Sources

Prefecture-level city is the primary administrative unit in China, so it is more meaningful to take this type of city as the research object compared with county-level city. Therefore, this paper observes 126 prefecture-level resource-based cities in China from 2010 to 2019. Among them, 14 prefecture-level resource-based cities were removed from consideration due to the unavailability of data. The classification of these cities in China is shown in Table A1 in Appendix A.

The data in this paper are from China City Statistical Yearbook, China Labor Statistical Yearbook, China Population and Employment Statistical Yearbook, China Statistical Yearbook and some prefecture-level city statistical bulletins. The missing data are supplemented by linear interpolation.

#### 4.1.2. Variables

(1)　Explained variable

In this paper, the green development of resource-based cities is regarded as a process of investing capital, labor and energy factors and producing economic benefits and ecological environmental impact. Referring to the research [4,38,55,56] and combining the theories of green development, an evaluation index system of GDE in resource-based cities was built, as shown in Table 1.

**Table 1.** Evaluation index system of GDE in resource-based cities.

| Indicators | Primary Indicator | Secondary Indicator | Unit |
|---|---|---|---|
| Input | Financial input | Investment in fixed assets | CNY 10 thousand |
| | Labor input | Number of employees in urban units at the end of the period | 10 thousand persons |
| | Technology input | Expenditure on science and technology | CNY 10 thousand |
| | Energy input 1 | Total water supply | 1 hundred thousand tons |
| | Energy input 2 | Total electricity consumption of the whole society | 10 thousand kilowatt-hours |
| Output | Economic output | Gross regional product (GDP) | CNY 10 thousand |
| | Comprehensive pollutant emissions | Urban sulfur dioxide emissions | Ton |
| | | Industrial smoke (powder) dust emissions | Ton |
| | | Industrial wastewater discharge | Ton |

Among them, capital investment is a necessary fundamental element in the process of green development. Unlike cases in which provinces are the research object, the relevant data of the capital stock index are not easy to obtain and are seriously lacking in the study of cities. Meanwhile, the difference in the selection of capital stock and depreciation rate in the base period seriously affects the calculation results of capital stock in the current period, resulting in a significant deviation between statistical results and the actual results [4]. Therefore, this paper takes investment in fixed assets as the secondary indicator of capital input [4,38]. In addition, in terms of the economic output, GDP is converted into real GDP

based on the year 2000. Pollutant emission is an undesirable output in the process of urban green development. This paper draws on the research of [39,57] and uses the principal component synthesis method to synthesize the emissions of three pollutants in secondary indicators into total pollutant emissions.

(2)　Explanatory variables

　　a.　Industrial structure upgrading (ISU): This paper selects GDP of the tertiary industry and the secondary industry as the index of ISU [32]:

$$ISU = \frac{GDP\ of\ the\ tertiary\ industry}{GDP\ of\ the\ secondary\ industry} \tag{1}$$

　　b.　Human capital structure upgrading (HCSU):

There are four models for measuring the degree of structural change: the change value, the lead coefficient, the entropy index and the Moore value of industrial structure. The last model can reflect the change in direction of each industry in the process of structural evolution and can reflect the process of industrial structure change in more detail with higher sensitivity. The essence of this is the vector angle method. Therefore, this paper refers to [6,48,58,59] and adopts the vector angle method to measure HCSU. The higher the value of the calculation result, the higher the degree of HCSU. The calculation process is as follows:

Step 1: According to their education level, the current employees are divided into five classes: below primary school (excluding primary school), primary school, middle school, high school, college or above. The proportion of human capital at five levels (the proportion of the current number of employed people in the total number of employed people) is successively taken as a component of the five-dimensional human capital space vector, i.e., $X_0 = (x_{0,1},\ x_{0,2},\ x_{0,3},\ x_{0,4},\ x_{0,5})$.

Step 2: We select the basic unit vector group $X_1 = (1,0,0,0,0)$, $X_2 = (0,1,0,0,0)$, $X_3 = (0,0,1,0,0)$, $X_4 = (0,0,0,1,0)$, $X_5 = (0,0,0,0,1)$ as the reference vector, and measure the angle $\theta_j (j = 1,2,3,4,5)$ between the space vector of human capital $X_0$ and them successively:

$$\theta_j = \arccos\left( \frac{\sum\limits_{i=1}^{5} (x_{j,i} \cdot x_{0,i})}{\left(\sum\limits_{i=1}^{5} x_{j,i}^2\right)^{\frac{1}{2}} \left(\sum\limits_{i=1}^{5} x_{0,i}^2\right)^{\frac{1}{2}}} \right) \tag{2}$$

where $x_{j,i}$ is the *i*-th component of $X_j (j = 1,2,3,4,5)$ and $x_{0,i}$ is the *i*-th component of $X_0$.

Step 3: We calculate HCSU:

$$HCSU = \sum_{j=1}^{5} (W_j \cdot \theta_j) \tag{3}$$

where $W_j$ is the weight of $\theta_j$. Based on the coefficient of variation method, $W_1$, $W_2$, $W_3$, $W_4$, $W_5$ are respectively set as 5, 4, 3, 2, 1 [6,58]. The higher the value of HCSU, the higher the level of human capital structure.

Step 4: At present, this method is only widely applied at provincial level, and relevant data on the educational level of the employed population at the city level are scarce. Therefore, we use the improved method proposed by [59]. The calculation formula of HCSU is as follows:

$$HCSU_{prefecture} = HCSU_{province} \times \frac{the\ number\ of\ higher\ education\ students\ of\ the\ prefecture}{the\ number\ of\ higher\ education\ students\ of\ the\ province} \tag{4}$$

(3)　Control variables

The control variables are listed in Table 2:

**Table 2.** Control variables.

| Variables | Symbol | Description |
|---|---|---|
| Government support | Gov | Local general public budget expenditure/Gross regional product |
| Scientific and technological innovation | Sci | Number of green patents |
| Basic infrastructure | Basic | Highway mileage/Area of land |
| Environmental regulation | Envi | Comprehensive indicators of environmental regulation (the harmless treatment rate of domestic waste, the treatment rate of urban domestic sewage and the comprehensive utilization rate of industrial solid waste were selected and calculated with the principal component synthesis method [60]) |

### 4.2. Model Construction

In this paper, SBM-Undesirable-DEA model was selected to estimate the GDE in resource-based cities in China, and the econometric model was used to analyze its influencing factors. The flow chart is shown in Figure 1.

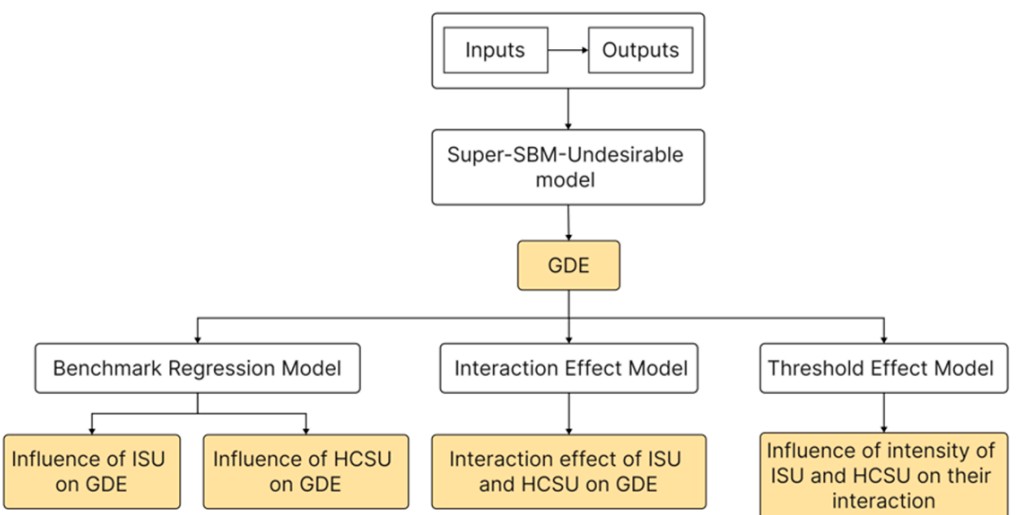

**Figure 1.** Flow chart of model construction.

### 4.2.1. Super-SBM-Undesirable Model

DEA as a nonparametric method is widely used in measuring efficiency in wide fields such as ecology, banking industry, etc. Traditional DEA models such as CCR and BCC are based on radial measurement and one-side orientation, indicating there may be a resulting bias. In other words, the results calculated by traditional DEA models are overestimated when there is a nonzero slack of inputs or outputs. Aiming to solve this problem, Tone proposed SBM model, which is based on slack variables instead of requiring proportional changes of inputs or outputs [61]. In addition, the model is easy to incorporate undesirable outputs into due to the slacks. However, there is another problem in traditional DEA model and also SBM model, in that efficient DMUs cannot be compared with each other because their efficiency values are all equal to 1. To solve this problem, this paper combined SBM-Undesirable model and super efficiency model [62], constructing Super-SBM-Undesirable model to calculate GDE in resource-based cities. In addition, due to the sulfur removal device, urban sulfur dioxide emissions can be increased "freely". Thus, this

undesirable output should be extended as strong disposable [63]. The same is true of the other undesirable outputs. On such basis, the equation can be written as

$$
\begin{aligned}
min \quad \rho &= \frac{1 - \frac{1}{m}\sum\limits_{i=1}^{m}\frac{s_i^-}{x_{ik}}}{1 + \frac{1}{q+s}\left(\sum\limits_{r=1}^{q}\frac{s_r^+}{y_{rk}} + \sum\limits_{t=1}^{s}\frac{s_t^-}{b_{tk}}\right)} \\
s.t. \quad & \sum\limits_{\substack{j=1 \\ j\neq k}}^{n} x_{ij}\lambda_j + s_i^- \leq x_{ik}, i = 1,\dots,m, \\
& \sum\limits_{\substack{j=1 \\ j\neq k}}^{n} y_{rj}\lambda_j - s_r^+ \geq y_{rk}, r = 1,\dots,q, \\
& \sum\limits_{\substack{j=1 \\ j\neq k}}^{n} b_{tj}\lambda_j + s_t^- \leq b_{tk}, t = 1,\dots,s, \\
& \sum\limits_{\substack{j=1 \\ j\neq k}}^{n} \lambda_j = 1, \\
& \lambda_j \geq 0, j = 1,\dots,n, \\
& s_i^-, s_r^+, s_t^- \geq 0, i = 1,\dots,m, r = 1,\dots,q, t = 1,\dots,s
\end{aligned}
\tag{5}
$$

where there are $n$ DMUs, and each DMU has $m$ inputs, $q$ desirable outputs and $s$ undesirable outputs. $s_i^-$, $s_r^+$, $s_t^-$ represent the slack variables. $x_{ij}, y_{rj}, b_{tj}$ represent the $i$-th input, the $r$-th desirable output and the $t$-th undesirable output of the $j$-th DMU, respectively. $\lambda_j$ is the combination coefficient of DMUs. It is worth mentioning that if the efficiency value $\rho \geq 1$, the evaluated DMU is efficient.

### 4.2.2. Econometric Model

**(1) The Benchmark Regression Model**

First, based on hypotheses 1 and 2, Models 1, 2 and 3 were constructed, respectively, to explore the impact of ISU and HCSU on the GDE in resource-based cities. The specific formulas are as follows:

$$
GDE_{it} = \alpha_1 + \beta_1 \ln ISU_{it} + \gamma_1 Control_{it} + \mu_i + \varepsilon_{it} \tag{6}
$$

$$
GDE_{it} = \alpha_2 + \beta_2 \ln HCSU_{it} + \gamma_2 Control_{it} + \mu_i + \varepsilon_{it} \tag{7}
$$

$$
GDE_{it} = \alpha_3 + \beta_{31} \ln ISU_{it} + \beta_{32} \ln HCSU_{it} + \gamma_3 Control_{it} + \mu_i + \varepsilon_{it} \tag{8}
$$

where, $i$ represents different cities, and $t$ represents different years. The explained variable $GDE_{it}$ is green development efficiency. $ISU_{it}$ is the upgrading level of industrial structure. $HCSU_{it}$ is the high-level index of human capital structure. To avoid heteroscedasticity, this paper took logarithm of $ISU_{it}$ and $HCSU_{it}$. $Control_{it}$ represents control variables, including $Cov_{it}$, $Sci_{it}$, $Baisc_{it}$ and $Envi_{it}$. $\mu_i$ represents the individual fixed effect. $\varepsilon_{it}$ represents the random disturbance term. $\alpha$ is the constant term. $\beta$ and $\gamma$ are the coefficient values. If the $p$-values of $\beta$ and $\gamma$ are smaller than 0.1, it means that the corresponding variable can significantly impact GDE.

**(2) Interaction Effect Model**

Based on Hypothesis 3-1, this paper further introduces the interaction terms of ISU and HCSU to build Model 4 as Formula (10), so as to examine the impact of their interaction effects on the GDE in resource-based cities.

$$
GDE_{it} = \alpha_3 + \beta_{41} \ln ISU_{it} + \beta_{42} \ln HCSU_{it} + \beta_{43} \ln ISU_{it} \times \ln HCSU_{it} + \gamma_4 Control_{it} + \mu_i + \varepsilon_{it} \tag{9}
$$

If the *p*-values of $\beta_{43}$ are smaller than 0.1, it means that the interaction between ISU and HCSU has a significant impact on urban GDE.

(3)   Threshold Effect Model

Based on Hypothesis 3-2, this paper takes HCSU and ISU as the threshold variables and analyzes the influence of their intensity on their interaction. The threshold effect refers to the following phenomenon: when the parameter of a variable is at a certain level, a sudden change in the other economic parameters can affect it. The critical value that produces this phenomenon is called the threshold value. Since the number of thresholds may not be unique, this paper constructs single- and double-threshold panel models, as shown in Equations (10) and (11), respectively:

$$GDE_{it} = \eta_0 + \eta_1 ISU_{it} + \eta_2 HCSU_{it} + \eta_3 ISU_{it} \times HCSU_{it} I(Adj_{it} \leq \gamma) + \eta_4 ISU_{it} \times HCSU_{it} I(Adj_{it} > \gamma) \\ + \lambda_5 Control_{it} + u_{it} \tag{10}$$

$$GDE_{it} = \lambda_0 + \lambda_1 ISU_{it} + \lambda_2 HCSU_{it} + \lambda_3 ISU_{it} \times HCSU_{it} I(Adj_{it} \leq \phi_1) \\ + \lambda_4 ISU_{it} \times HCSU_{it} I(\phi_1 < Adj_{it} < \phi_2) + \lambda_5 ISU_{it} \times HCSU_{it} I(Adj_{it} \geq \phi_2) + \zeta_6 Control_{it} + u_{it} \tag{11}$$

where $Adj_{it}$ represents the threshold variable; $I(\cdot)$ is the indicative function; $\gamma$, $\varphi_1$ and $\varphi_2$ are the threshold values; $\mu_{it}$ is the random error term. If the *p*-value of the bootstrap self-sampling test is smaller than 0.1 and the estimations between $\eta_3$ and $\eta_4$ or between $\lambda_3$, $\lambda_4$ and $\lambda_5$ are different, it means that threshold effect and nonlinear relationship exist between the interaction term and GDE.

## 5. Empirical Results of GDE in China's Resource-Based Cities and Its Influencing Factors

*5.1. Analysis of GDE in China's Resource-Based Cities*

In this paper, the GDE in resource-based cities was measured using Model 5. The calculated GDEs during 2010–2019 are listed in Table A1 in Appendix A. Figure 2 shows the change in average GDE from 2010 to 2019. As can be seen in Figure 2, GDE showed a slow upward trend in fluctuations during 2010–2019, reaching a peak in 2019 at nearly 0.9. However, the overall GDE was relatively low in this period, at no more than 0.9. This suggests that although the green development performance in China's 112 resource-based cities has been improved, there is still room for improvements in efficiency.

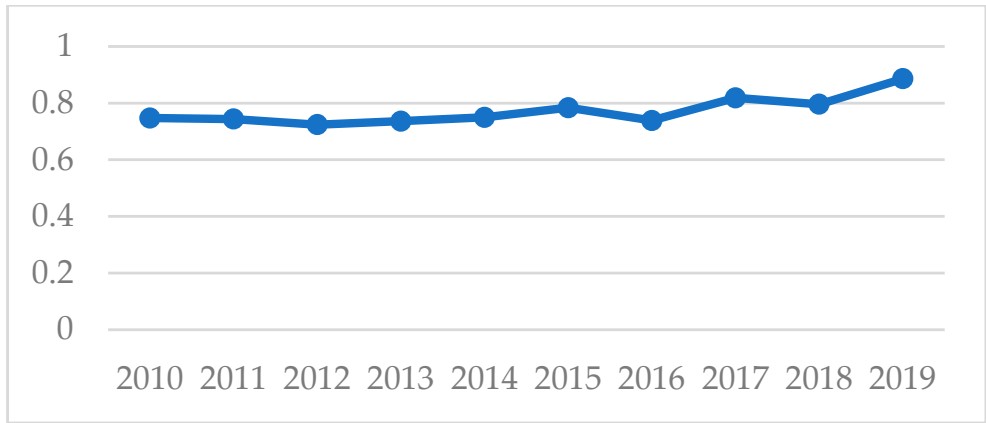

**Figure 2.** Average GDE in 112 resource-based cities in China from 2010 to 2019.

To visually observe the distribution and dynamic evolution of GDE, this paper further created a kernel density diagram, as seen in Figure 3. The graphical presentations include the GDE in 2010, 2015 and 2019. It can be seen that GDE distribution presents a single peak. In 2010, most GDEs concentrated on 0.7, and many cities have not reached an efficient frontier. It is worth noting that from 2010 to 2019, the peak tends to shift right and lower, meaning GDE increased in most cities and the development structure of China was consistently adjusted and optimized. This also suggests that GDE is influenced by ISU and

HCSU to a certain extent. The year 2019 exhibits a flat curve, with GDE gradually dispersed among cities, which can be related to the different conversions of input to corresponding pollutant discharges and economic benefits in different cities.

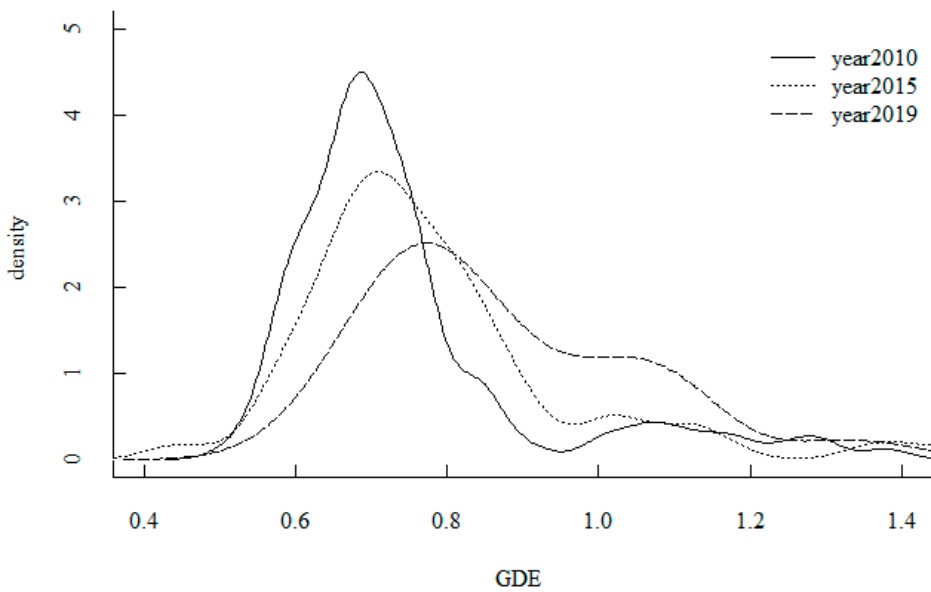

**Figure 3.** Kernel density diagram and dynamic evolution of GDE in resource-based cities in China, 2010–2019.

Overall, although green development in most resource-based cities in China is on track for steady improvement, it still has some room for improvement.

*5.2. Analysis of Influencing Factors of GDE in China's Resource-Based Cities*

This paper further explores whether ISU, HCSU and their interaction can improve GDE in resource-based cities. Firstly, as the Hausman test results of Models 1–4 all reject the null hypothesis, the fixed effects model was chosen. The specific results of the regression analysis are shown in Table 3.

The regression results of Models 1, 3 and 4 show that the regression coefficient of lnISU is significantly positive at the level of 1%. This indicates that on the basis of controlling the urban individual effect and other factors, ISU has a significant positive impact on GDE in resource-based cities. Hypothesis 1 is proved.

The regression results of Model 2 show that the regression coefficient of lnHSCU is 0.018, which is significant at the 1% level. It shows that when other variables are controlled unchanged, a 1% increase in the level of HCSU will increase the GDE by 0.018 on average. In other words, HCSU has a positive effect on the green development efficiency of resource-based cities, which is consistent with Model 3 and Model 4. Hypothesis 2 is proved.

Expanding to Model 3, it is worth noting that after the lnISU and lnHCSU are included together, the coefficient of lnHCSU decreases by 0.002. This indicates that the influence of HCSU on the GDE can play a role through the internal mechanism of ISU, which verifies Hypothesis 3-1.

In Model 4, on the one hand, GDE is positively promoted by ISU and HCSU. On the other hand, it is positively encouraged by the interaction effects of the two. This shows that during the sample study period, the ISU and HCSU of resource-based cities were generally matched, and the two played a synergistic role in promoting the green development of resource-based cities. Hypothesis 3-1 is verified.

The control variables of Model 4 were further analyzed. The coefficient of Gov, Sci and Basic is positive and significant, which means that government support, scientific and technological innovation and basic infrastructure play active roles in urban green

development. The reason for this is that global attention to green development and the unique national system in China have determined that the Chinese government must play active planning, guidance and support roles in GDE. In addition, improvements in scientific and technological innovation can promote the development of green production equipment. In addition, these improvements can improve production efficiency and the resource utilization rate, thus promoting GDE. The improvement of basic infrastructure can boost efficiency and the convenience of resource allocation through network effects and space–time compression, so as to better promote GDE. However, the coefficient of Envi is significantly negative. The reasons for this may be as follows: To realize energy conservation and emission reduction, enterprises are required to use green production equipment and monitor pollutants. Such measures can bring benefits to sustainable development in the long run. However, in the short run, enterprises face enormous production costs, producing the "crowding-out effect" on essential resources for green development, which eventually inhibits the GDE.

**Table 3.** Benchmark regression results.

| Variables | GDE | | | |
|---|---|---|---|---|
| | **Model 1** | **Model 2** | **Model 3** | **Model 4** |
| lnISU | 0.053 *** | | 0.053 *** | 0.054 *** |
| | (0.011) | | (0.011) | (0.011) |
| lnHSCU | | 0.018 *** | 0.016 ** | 0.016 ** |
| | | (0.007) | (0.007) | (0.007) |
| lnISU × lnHSCU | | | | 0.015 * |
| | | | | (0.009) |
| Gov | 0.020 | 0.043 *** | 0.022 * | 0.023 * |
| | (0.014) | (0.013) | (0.014) | (0.014) |
| Sci | 0.001 | 0.007 *** | 0.001 * | 0.001 * |
| | (0.001) | (0.001) | (0.001) | (0.003) |
| Basic | 0.240 *** | 0.291 *** | 0.231 *** | 0.234 *** |
| | (0.081) | (0.080) | (0.081) | (0.080) |
| Envi | −0.060 ** | −0.070 *** | −0.062 *** | −0.058 ** |
| | (0.024) | (0.024) | (0.024) | (0.024) |
| _cons | 0.793 *** | 0.784 *** | 0.808 *** | 0.805 *** |
| | (0.022) | (0.023) | (0.023) | (0.023) |
| $R^2$ | 0.058 | 0.067 | 0.065 | 0.059 |
| City | Yes | Yes | Yes | Yes |
| Year | No | No | No | No |
| Panel Models | Fe | Fe | Fe | Fe |

Note: ***, ** and * indicate significant levels of 1%, 5% and 10%, respectively.

### 5.3. Panel Threshold Effect Analysis

The intensity of ISU and HCSU may affect their matching degree. In order to explore the nonlinear influence of their interaction effect on the GDE in resource-based cities, this paper took them as threshold variables, building the Hansen threshold model. Firstly, we determined whether a threshold effect existed, using the threshold effect test shown in Table 4.

**Table 4.** Threshold effect test.

| Threshold Variable | Model | Threshold Value | F-Value | *p*-Value | Bootstrap |
|---|---|---|---|---|---|
| ISU | Single-threshold | 1.9726 | 22.97 | 0.05 | 300 |
| | Double-threshold | [0.2899, 1.9726] | 7.03 | 0.433 | 300 |
| HCSU | Single-threshold | 0.0963 | 14.20 | 0.437 | 300 |
| | Double-threshold | [0.0963, 1.7872] | 13.25 | 0.260 | 300 |

Combined with Table 4, it can be seen that ISU only passes the single-threshold test, and the threshold value is 1.9726. However, it fails the double-threshold test. This indicates that when ISU is taken as the threshold variable, the interaction term of HCSU and ISU has a single-threshold effect on the GDE. The single- and double-threshold tests of HCSU fail, that is, because when HCSU is taken as the threshold variable, there is no threshold effect. Therefore, this paper took ISU as the threshold variable to build a single-threshold model, and the results are shown in Table 5.

**Table 5.** Regression results of threshold model.

| Variables | Coef. | SE | *p*-Value |
|---|---|---|---|
| $ISU_{it} \times HCSU_{it}$ ($ISU_{it} \leq 1.9726$) | 0.031 *** | 0.007 | 0.038 |
| $ISU_{it} \times HCSU_{it}$ ($ISU_{it} > 1.9726$) | −0.029 *** | 0.008 | 0.003 |
| N | | 1120 | |

Note: *** indicates significant levels of 1%.

It can be seen that when the level of ISU is on the left of the threshold value (1.9726), the interaction term is significantly positive at the 1% level. This indicates that ISU and HCSU match each other well in this case. Their supply and demand reach an optimal level. ISU breeds many new high-tech industries and releases the demand for a large amount of high-quality labor [64]. At the same time, HCSU can continuously provide intellectual and technical support for the gap in high-quality talents caused by ISU [64]. In addition, it can increase the demand for knowledge-intensive products. When the intensity of the two is increased, it not only enhances their own economic and ecological benefits, but also makes the other side play a more significant role in promoting GDE. Hypothesis 3-2 is verified.

When the level of ISU is on the right side of the threshold value (1.9726), the interaction presents interference. The reasons for this are that the resource-based cities in China highly depend on natural resources, there is a lack of a talent-gathering advantage, and their talent policy is deficient. When ISU is too advanced, the human capital may not be able to meet the configuration requirements required by ISU. Therefore, ISU appears to hollow out, which reduces the synergy of ISU and HCSU. Eventually, their interaction can inhibit GDE. Hypothesis 3-2 is verified.

*5.4. Robustness Test*

In order to ensure the reliability of the research results, this paper tested their robustness by transforming the explanatory variables method, instrumental variable method and random sample deletion method.

5.4.1. Transform Explanatory Variables Method

This paper replaced the original ISU indicator to carry out the robustness test. According to [65,66], the new measure method of the ISU variable is shown in Formula (12):

$$NewISU = \sum_{i=1}^{3} (I_i \times i) = I_1 \times 1 + I_2 \times 2 + I_3 \times 3 \tag{12}$$

where $I_i$ is the proportion of *i*-th (*i* = 1, 2, 3) industry output in the GDP.

The regression results are shown in Table 6. The significance and direction of coefficients in this model are consistent with the original model. Therefore, the robustness of the benchmark regression results is confirmed.

**Table 6.** Robustness test results obtained by transforming explanatory variables.

| Explanatory Variable | GDE | | | |
|---|---|---|---|---|
| | **Model 1** | **Model 2** | **Model 3** | **Model 4** |
| lnNewISU | 0.086 *** | | 0.085 *** | 0.122 *** |
| | (0.024) | | (0.024) | (0.026) |
| lnHCSU | | 0.018 *** | 0.017 ** | 0.019 *** |
| | | (0.007) | (0.007) | (0.007) |
| lnNewISU × lnHCSU | | | | 0.075 *** |
| | | | | (0.022) |
| Gov | 0.329 ** | 0.043 *** | 0.035 *** | 0.033 ** |
| | (0.013) | (0.013) | (0.013) | (0.013) |
| Sci | 0.001 ** | 0.007 *** | 0.001 *** | 0.001 ** |
| | (0.001) | (0.001) | (0.001) | (0.001) |
| Basic | 0.282 *** | 0.291 *** | 0.272 ** | 0.272 *** |
| | (0.080) | (0.080) | (0.080) | (0.080) |
| Envi | −0.063 *** | −0.070 *** | −0.065 *** | −0.066 *** |
| | (0.024) | (0.024) | (0.024) | (0.024) |
| _cons | 0.305 *** | 0.784 *** | 0.325 *** | 0.126 *** |
| | (0.024) | (0.023) | (0.130) | (0.142) |
| R2 | 0.064 | 0.074 | 0.071 | 0.062 |
| City | Yes | Yes | Yes | Yes |
| Year | No | No | No | No |
| Panel Models | Fe | Fe | Fe | Fe |

Note: *** and ** indicate significant levels of 1% and 5%, respectively.

5.4.2. Random Sample Deletion Method

In this paper, the data from 2012 and 2017 were randomly eliminated. The new regression results are shown in Table 7. In Models 1–4, the regression results are consistent with the original ones. The robustness test based on the variable elimination method was passed.

**Table 7.** Robustness test results obtained by excluding 2012 and 2017.

| Explanatory Variable | GDE | | | |
|---|---|---|---|---|
| | **Model 1** | **Model 2** | **Model 3** | **Model 4** |
| lnISU | 0.061 *** | | 0.060 *** | 0.062 *** |
| | (0.017) | | (0.017) | (0.017) |
| lnHCSU | | 0.014 ** | 0.012 ** | 0.012 * |
| | | (0.008) | (0.007) | (0.007) |
| lnISU × lnHCSU | | | | 0.033 *** |
| | | | | (0.011) |
| Gov | −0.016 | 0.226 ** | 0.013 | 0.090 |
| | (0.109) | (0.094) | (0.110) | (0.113) |
| Sci | 0.001 | 0.001 | 0.001 | 0.001 |
| | (0.001) | (0.001) | (0.001) | (0.001) |
| Basic | 0.258 *** | 0.302 *** | 0.249 *** | 0.245 ** |
| | (0.095) | (0.094) | (0.095) | (0.094) |
| Envi | −0.061 ** | −0.068 ** | −0.062 ** | −0.057 ** |
| | (0.028) | (0.029) | (0.028) | (0.028) |
| _cons | 0.805 *** | 0.745 *** | 0.812 *** | 0.794 *** |
| | (0.036) | (0.031) | (0.036) | (0.036) |
| R2 | 0.056 | 0.062 | 0.061 | 0.053 |
| City | Yes | Yes | Yes | Yes |
| Year | No | No | No | No |
| Panel Models | Fe | Fe | Fe | Fe |

Note: ***, ** and * indicate significant levels of 1%, 5% and 10%, respectively.

### 5.4.3. Robustness Test of the Threshold Effect—Random Sample Deletion Method

This paper also used the method of randomly deleting the sample to analyze the robustness of the threshold model. The regression results after excluding 112 samples from 2010 are shown in Table 8. It can be seen that the threshold model is still significant, and the direction of action remains unchanged, passing the robustness test.

**Table 8.** Robustness test results of the threshold effect.

| Variables | Coef. | SE | *p*-Value |
| --- | --- | --- | --- |
| $HCSU_{it} \times ISU_{it}$ $(ISCU_{it} \leq 1.9578)$ | 0.028 ** | 0.015 | 0.069 |
| $HCSU_{it} \times ISU_{it}$ $(Indu_{it} > 1.9578)$ | −0.032 *** | 0.009 | 0.001 |
| Threshold estimate | | 1.9578 | |
| Number of periods | | 9 | |
| Number of cities | | 112 | |

Note: *** and ** indicate significant levels of 1% and 5%, respectively.

### 5.5. Heterogeneity Analysis

According to the resource exploitation level and urban development level, resource-based cities are divided into four types by the Plan: mature, growing, declining and regenerating cities. Their foundations and driving forces for green development are different. Therefore, this paper further analyzes heterogeneity based on the resource-based city types.

#### 5.5.1. Heterogeneity Analysis of GDE in Resource-Based Cities

Taking 2019 as an example, the proportion of the number of efficient cities in the total number of resource-based cities is shown in Figure 4.

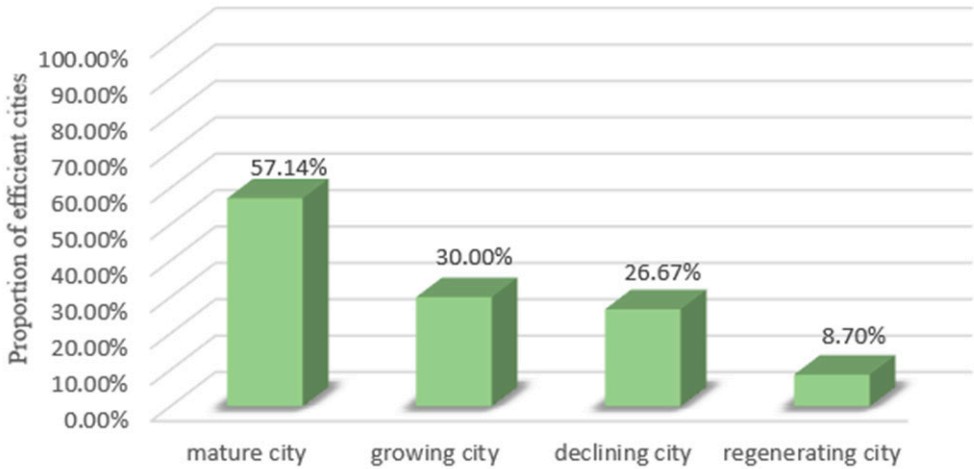

**Figure 4.** Proportion of efficient cities in 2019.

Among the growing cities, 57.14% of the cities had GDE values greater than or equal to 1 in 2019, ranking first and far higher than other types of cities. In the mature cities, 30.00% of the cities reached a DEA efficient state, ranking second. The reason for the large number of efficient cities in two types of cities is that the resource base is relatively strong, and it is easier to form scale advantages. Moreover, growing cities are more ecologically recoverable, and more flexible in the ways they can transition to green development. It is easier to achieve a win–win situation for economic development, resource conservation and environmental protection.

In the declining and regenerating cities, the number of efficient cities is relatively low. The reasons are as follows. Currently, most declining cities are in a state of "depleted

mines and declining cities", with significant characteristics of crude economic development and a situation of "pollution for growth". These cities need to dispose of the traditional economic development model and seek new momentum for green development. As for regenerating cities, although they no longer rely on single resource supplies, their green industry foundations are weak. It is difficult for them to generate high GDE.

5.5.2. Heterogeneity Analysis of Influencing Factors of GDE in Resource-Based Cities

The heterogeneity analysis results are shown in Table 9.

**Table 9.** Heterogeneity analysis results.

| Explanatory Variable | GDE | | | |
| --- | --- | --- | --- | --- |
| | (1) Mature City | (2) Growing City | (3) Declining City | (4) Regenerating City |
| lnISU | 0.069 *** | −0.231 | 0.086 ** | 0.089 *** |
| | (0.087) | (0.041) | (0.055) | (0.064) |
| lnHCSU | 0.019 * | 0.022 *** | 0.023 ** | 0.010 |
| | (0.007) | (0.005) | (0.005) | (0.006) |
| lnISU × lnHCSU | 0.006 * | 0.036 | 0.030 ** | −0.001 |
| | (0.038) | (0.056) | (0.062) | (0.063) |
| Gov | 0.249 * | −0.028 | 0.197 *** | 0.292 |
| | (0.059) | (0.046) | (0.069) | (0.054) |
| Sci | −0.001 | 0.004 * | 0.002 ** | 0.001 *** |
| | (1.576) | (1.076) | (1.015) | (1.065) |
| Basic | 0.166 | 2.478 ** | 0.266 *** | 0.106 |
| | (0.28) | (0.403) | (0.523) | (0.812) |
| Envi | −0.034 | −0.147 * | −0.062 | −0.776 * |
| | (0.624) | (0.609) | (0.505) | (1.69) |
| _cons | 0.804 *** | 0.992 *** | 0.666 *** | 0.783 *** |
| | (0.045) | (0.024) | (0.018) | (0.047) |
| City | Yes | Yes | Yes | Yes |
| Year | No | No | No | No |
| Panel Models | Fe | Fe | Fe | Fe |

Note: ***, ** and * indicate significant levels of 1%, 5% and 10%, respectively.

(1)　Mature cities

Column 1 shows that ISU, HCSU and their interaction have significantly improved the GDE in mature cities, which is consistent with the overall results. Mature cities have relatively strong foundations of human resources, technical levels and natural resources. In addition, residents' environmental awareness tends to be strong. Cooperation between human capital and industrial structure is more mature, and more likely to play a positive role in promoting green development.

(2)　Growing cities

As we can see from Column 2, HCSU significantly improves the GDE of growing cities, but ISU and the interaction between them have no significant impact on the GDE. The reasons for this are as follows: Most growth-resource-based cities are located in the Western region. Although the Chinese government has been supporting construction in the Western region, and local resources are abundant, ISU cannot currently be translated into economic value. This may be due to the region's limited local geographical location, poor economic foundations, out-of-date production technology and lower levels of environmental investment. The industrial structure development direction is unclear, and the upgrading speed is slow. The demand for industrial structure cannot match the supply of existing human capital.

(3)　Declining cities

Column 3 shows that ISU, HCSU and their interaction can promote the GDE in declining cities. Specifically, firstly, the condition of industrial structure in declining cities is serious, and the proportion of heavy industry is too high. These cities need to upgrade their industrial structures to promote green transformation. Secondly, declining cities in China are mostly located in the Eastern and Central regions, with relatively good levels of urban economic development and openness. The government's support of and investment in science and technology education are great. In recent years, most cities have introduced various policies to attract high-quality talent, leading to high-quality human capital accumulation. Thus, this has played a vital role in promoting green economic development. ISU and HCSU can encourage each other in the transformation and upgrading processes of such cities, meaning they can have a positive interaction.

(4)    Regenerating cities

In Column 4, ISU positively promotes the GDE of regenerating cities, but HCSU and the interaction between the two are insignificant. Most regenerating cities in China are located in Shandong, Jiangsu, and three Eastern provinces. They are facing problems such as unreasonable investment structure, insufficient extension length of the industrial chain and immature regional cooperation mechanisms. Urban industrial transformation can help these cities get rid of the "resource curse". However, due to the lack of experience, human capital training cannot fully match the actual demand of the industry and wastes some resource investment opportunities. It is still necessary to explore an effective development mode.

Based on the above heterogeneity analysis, Hypothesis 4 is verified.

## 6. Discussion and Conclusions

### 6.1. Discussion

This paper takes 112 resource-based cities in China as its research object. We measured their GDE and analyzed its influencing factors. The research results can help the government and relevant enterprises implement more precise measures for the green transformation of different types of resource-based cities. The research ideas and methods used in this paper are not only applicable to the analysis of resource-based cities but can also be extended to other countries and industries.

It was found that GDE varies greatly among cities, and the number of efficient cities shows a trend of first decreasing and then fluctuating and increasing, which is consistent with the research of [9] for oil- and gas-resource-based cities.

In addition, most existing studies on the impact of ISU and HCSU on GDE are based on the national scope, and they do not discuss the urgency of the green development transformation of resource-based cities. This paper makes improvements on this issue and finds that the upgrading of HCSU can improve the GDE of resource-based cities, which is consistent with the results based on the national level [6,48]. However, this conclusion is contrary to [26]. This may be due to covering different samples and different periods. Moreover, we conclude that ISU can help improve GDE, which is consistent with existing research [8,10,25,67], although their research objects are Vietnam, China's provinces or the lower reaches of the Yellow River Basin.

The mediating effect or moderating effect transmission path of "human capital structure upgrading → industrial structure upgrading → sustainable economic development" and the influencing path of "industrial structure upgrading → human capital structure upgrading" have been confirmed by scholars [6,27,46,47]. As is shown in this paper, we believe that the two paths both exist, and they have mutual influence (interaction effect) on GDE in resource-based cities; these hypotheses are verified in this paper. In addition, this paper takes industrial structure upgrading as the threshold variable in order to analyze the threshold effect of their interaction, which fills the research gap on this subject.

Resource-based cities are divided into four types: mature, growing, declining and regenerating cities. Their foundations and driving forces of green development vary greatly, and we recognize the necessity of this. We conclude that the GDE of resource-based

cities presents the characteristics of "growing cities > mature cities > regenerating cities > declining cities", which is roughly the same as the findings for resource-based cities in arid areas and the Yellow River Basin [68,69]. However, it was found that the average order of the industrial green transformation efficiency of resource-based cities was as follows: "declining cities > regenerating cities > growing cities > mature cities" [70]. The reason for this may be this paper's different perspective. Specifically, this paper focuses on static perspectives, while other research has used the GML (global Malmquist–Luenberger) productivity index from a dynamic perspective and only focused on the industrial sector. Thus, the static efficiency of regenerating and declining cities is low since they are in a state of resource exhaustion or slow economic development. Nevertheless, they actively seek green transformation development, which has begun to bear fruit, and they have obtained a high dynamic efficiency. Comparatively speaking, the transformation speed of the growing cities is relatively stable because they have sufficient resources and better environmental protection. The development mode of mature cities is more stable and efficient, but there is a bottleneck in transformation.

In addition, the effects of ISU and HCSU on the GDE of different resource-based cities are also heterogeneous. Each type of resource-based city can find referential urban development cases in the global scope according to its own development stage. We can take Anshan City (a regenerating city in China) as an example. By referring to the German "dual system" vocational education mode, it initially formed a "dual cultivation system" for technical talents which included "government guidance, enterprise leading, platform support and mechanism innovation". In 2021, the ratio of primary, secondary and industrial added values in Anshan was 6.4:41.8:51.8, meaning that the effect was remarkable. Thus, other old industrial cities and resource-based cities can learn from this example. According to [5], Houston's industrial chain extension and intensive development model, Bakupei's diversified new-type leading industry development model, Kyushu's government assistance to guide low-carbon industry implantation development model and Birmingham's cultivation of non-resource-based industry leading development model can all provide reference for the other resource-based cities.

*6.2. Conclusions*

Based on the panel data of 112 resource-based cities in China during 2010–2019, this paper uses the SBM-Undesirable-DEA model to calculate the GDE of various cities. Furthermore, the dynamic evolution and restraining factor of GDE are explored. On such basis, this paper expounds the ways ISU and HCSU influence GDE from a theoretical point of view. In addition, the fixed effect model is used for verification. In addition, the interaction effect model and threshold effect model are used to explore their matching degree. Finally, the heterogeneity based on four types of resource-based cities, namely mature, growing, declining and regenerating, is explored.

The main conclusions are as follows:

(1) In general, from 2010 to 2019, GDE showed a steady development trend, but there was still some room for improvement. The results of the redundancy analysis of GDE show that redundant pollutant emission is still the most important factor restricting the green development of resource-based cities.

(2) ISU and HSCU can directly promote the GDE in resource-based cities. With the increasing intensity of ISU, the matching degree of the two factors influencing the GDE in resource-based cities decreases. When the level of ISU is lower than the threshold value, the two match each other and generate "additional" force to significantly promote GDE. When ISU exceeds the threshold value, the matching effect between the two decreases.

(3) The value of GDE and the effects of ISU and HSCU on GDE are different across four types of resource-based cities.

**7. Policy Recommendations, Limitations and Future Studies**

*7.1. Policy Recommendations*

Based on the research on the GDE in resource-based cities, this paper gives the following three suggestions.

(1) Resource allocation should be optimized. The government should increase investment in environmental protection and urge enterprises to take responsibility for ESG (environment, social and governance). Enterprises should improve energy efficiency by upgrading low-carbon technology, utilizing low-carbon products, and using clean energy. For example, they can introduce foreign advanced desulfurization and denitrification technology. In addition, measures such as increasing carbon quotas, carbon trading and carbon finance can be adopted.

(2) HUSU and ISU should be promoted simultaneously. To solve the "bottleneck" of technology, the government should increase expenditure on talent training, and improve the talent introduction policy. In addition, a reproducible experience for the revitalization of resource-based cities can be formed. For example, these cities can learn from Guizhou and Shandong provinces in China to set up industrial technology research institutions to attract excellent enterprises and teams. Therefore, human capital will be easier to match with the local industrial structure, so as to produce the effect of "1 + 1 > 2". It should also be borne in mind that the industrial structure cannot develop excessively.

(3) Resource-based cities should construct different green and low-carbon transformation paths based on city types. Specifically, for mature cities, they should build new high-tech replacement industries, develop "production + living" service industries and cultivate the diversified transformation of existing industries. For growing cities, they should make complete preparations before resource depletion. They can focus on guiding industrial structure and human resources to match. In addition, they can develop cloud computing, blockchain and other information technologies to explore green and smart industries systems. For declining cities, they should make the best of their regional and talent advantages, improve social security, encourage innovation and support the establishment of low-carbon alternative industries. Among the four resource-based city types, the regenerating city is the healthiest type. These cities can make full use of existing industrial structure advantages, cultivate matching labor, pay attention to cultural construction, extend the industrial chain length and promote the coordinated and cooperative development of regions. Finally, the government should support the establishment of a demonstration zone and regional central cities.

*7.2. Limitations and Future Studies*

This paper takes 112 resource-based cities in China as its research object. We have measured their GDE and analyzed its influencing factors. The research results can help the government and relevant enterprises implement more precise measures for the green transformation of different types of resource-based cities. The research ideas and methods used in this paper are not only applicable to the analysis of resource-based cities but also can be extended to other countries and industries. However, there are still some limitations worth discussing. First, the data in this paper are only fully updated to 2019 since the data after 2019 are not retrievable with current data sources. The authors of this paper will conduct further comparative research as data are updated. Second, this paper does not explore the impact of overall HCSU, which combines the quality and quantity of human capital, on green development. This can be further improved. Third, the accuracy of this paper can be improved. The R2 of the above models is minor, and may be related to insufficient selection of control variables. The selection of different control variables has different effects on and significance for the empirical results. The control variables can be further supplemented and improved. Moreover, heteroscedasticity, endogeneity and other problems are difficult to solve entirely.

For further research, other performance evaluations of the green development of resource-based cities can be examined. Furthermore, other influencing factors of the GDE in resource-based cities can also be explored, such as FDI, which has been proved to be a significant driving force in China's urbanization [71], wellbeing, safety, health and government governance. Government governance can guide the flow of resources. It is important to promote the development of ecofriendly enterprises and contribute to the upgrading of industrial structure, which will improve the green development of resource-based cities. This can be regarded as a future research direction. In addition, after the data are updated, the updated results can be calculated.

**Author Contributions:** Conceptualization, W.S., Y.L., X.L. and J.S.; methodology, Y.L., X.L. and J.S.; software, Y.L., X.L. and J.S.; validation, Y.L.; formal analysis, X.L. and J.S.; investigation, W.S., Y.L. and X.L.; resources, W.S., Y.L., W.L. and C.L.; data curation, X.L.; writing—original draft preparation, W.S., Y.L. and J.S.; writing—review and editing, W.S., Y.L., J.S. and C.L.; visualization, X.L. and J.S.; supervision, W.S., W.L. and C.L.; project administration, W.S., W.L. and C.L.; funding acquisition, W.S. and W.L.. All authors have read and agreed to the published version of the manuscript.

**Funding:** This research was funded by the National Natural Science Foundation of China (NSFC), grant numbers 11971259 and 12171042; the National Social Science Foundation of China (NSSFC), grant number 21BTJ072; and the Key R&D Plan of Shandong Province (Soft Science Project, Key Project), grant number 2022RZB01005.

**Institutional Review Board Statement:** Not applicable.

**Informed Consent Statement:** Not applicable.

**Data Availability Statement:** The data are not publicly available due to privacy restrictions.

**Acknowledgments:** This study was sponsored by the National Natural Science Foundation of China (NSFC: 11971259) and the Key R&D Plan of Shandong Province (Soft Science Project, Key Project) (grant number: 2022RZB01005), which is highly acknowledged.

**Conflicts of Interest:** The authors declare no conflict of interest.

## Abbreviations

| | |
|---|---|
| GDE | green development efficiency |
| ISU | industrial structure upgrading |
| HCSU | human capital structure upgrading |
| SDGs | sustainable development goals |
| UNDP | United Nations Development Programme |
| GDP | Gross regional product |

## Appendix A

**Table A1.** Classification of resource-based cities.

| Growing cities |
|---|
| Prefecture-level administrative regions: Shuozhou City, Hulunbuir City, Ordos City, Songyuan City, Hezhou City, Nanchong City, Liupanshui City, Bijie City, Qiannan Buyei and Miao Autonomous Prefecture, Qiandinan Buyei and Miao Autonomous Prefecture, Zhaotong City, Chuxiong Dance Autonomous Prefecture, Yan 'an City, Xianyang City, Yulin City, Wuwei City, Qingyang City, Longnan City, Haixi Mongolian and Tibetan Autonomous Prefecture, Altay Region; County-level cities: Holingol City, Xilinhot City, Yongcheng City, Yuzhou City, Lingwu City, Hami City, Fukang City; Counties: Yingshang County, Dongshan County, Changle County, Shan County. |

**Table A1.** *Cont.*

| Mature cities |
|---|
| Prefecture-level administrative districts: Zhangjiakou City, Chengde City, Xingtai City, Handan City, Datong City, Yangquan City, Changzhi City, Jincheng City, Xinzhou City, Jinzhong City, Linfen City, Yuncheng City, Luliang City, Chifeng City, Benxi City, Jilin City, Yanbian Korean Autonomous Prefecture, Heihe City, Daqing City, Jixi City, Mudanjiang City, Huzhou City, Suzhou City, Bozhou City, Huainan City, Chuzhou City, Chizhou City, Xuancheng City, South Ping City, Sanming City, Longyan City, Ganzhou City, Yichun City, Dongying City, Jining City, Tai 'an City, Laiwu City, Sanmenxia City, Hebi City, Pingdingshan City, Ezhou City, Hengyang City, Chenzhou City, Shaoyang City, Loudi City, Yunfu City, Baise City, Hechi City, Guangyuan City, Guang 'an City, Zigong City, Panzhihua City, Dazhou City, Ya 'an City, Liangshan Dance Autonomous Prefecture, Anshun City, Qujing City, Baoshan City, Pu 'er City, Lincang City, Weinan City, Baoji City, Jinchang City, Pingliang City, Karamay City, Bayingoleng Mongolian Autonomous Prefecture; County-level cities: Luquan City, Renqiu, Gujiao City, the mountain city, phoenix city, Shangzhi, Chaohu, Longhai City, Ruichang City, Guihou City, Derong City, Zhaodan City, Pingdu city, Dengke Vity, Xinrong City, Gongyi City, Congyang City, Yingpei City, Yidu City, Liuyang City, Linxiang City, Gaoyao City, Cenxi City, Dongfang, Mianzhu City, Qingzhen, Anning City, Kaiyuan, Hetian City; Counties/autonomous counties/forest areas: Qinglong Manchu Autonomous County, Yi County, Liangyuan County, Quyang County, Kuandian Manchu Autonomous County, Yi County, Wuyi County, Qingtian County, Pingtan County, Xingzi County, Wannian County, Baokang County, Shennongjia Forest District, Ningxiang County, Taojiang County, Huayuan County, Lianping County, Long 'an County, Longsheng Autonomous County, Teng County, Xiangzhou County, Qiongzhong Li and Miao Autonomous County, Lingshui Li Autonomous County, Ledong Li Autonomous County, Tongliang County, Rongchang County, Dianjiang County, Chengkou County, Fengjie County, Xiushan Tujia and Miao Autonomous County, Xingwen County, Kaiyang County, Xiuwen County, Zunyi County, Song County Peach and Miao Autonomous Counties, Jining County, Xinping and Dai Autonomous Counties, Lanping Bai Pumi Autonomous Counties, Makuan County, Qusong County, Qingyang County, Luonan County, Maqu County, Datong Hui and Tu Autonomous Counties, Zhongning County, Baicheng County. |

| Declining cities |
|---|
| Prefecture-level administrative regions: Wuhai City, Fuxin City, Fushun City, Liaoyuan City, Baishan City, Yichun City, Hegang City, Shuangyashan City, Qitaihe City, Greater Hinggan Mountains Region, Huaibei City, Tongling City, Jingdezhen City, Xinyu City, Pingxiang City, Zaozhuang City, Jiaozuo City, Puyang City, Huangshi City, Shaoguan City, Luzhou City, Tongchuan City, Baiyin City, Shizuishan City; County-level cities: Huozhou City, Arshan City, Beitao City, Jiutai City, Shulan City, Dunhua City, Wudalianchi City, Xintai City, Lingbao City, Zhongxiang City, Daye City, Songzi City, Qianjiang City, Changning City, Leiyang City, Zixing City, Lengshuijiang City, Lianyuan City, Heshan City, Huachin City, Gejiu City, Yulen City; Counties/autonomous counties: Wangqing County, Dayu County, Changjiang Li Autonomous County, Yimen County, Tongguan County; Municipal districts/development and administrative districts: Jinglong Mining District, Xihuayuan District, Yingshuanyingzi Mining District, Shiguai District, Gongchangling District, Nanpiao District, Yangjiazhangzi Development Zone, ErDaojiang District, Jiawang District, Zichuan District, Pinggui Administrative District, Nanchuan District, Wansheng Economic Development Zone, Wanshan District, Dongchuan District, Honggu District. |

| Regenerating cities |
|---|
| Prefectural administrative regions: Tangshan, Baotou, Anshan, Panjin, Huludao Tonghua, Xuzhou, Suqian, Ma 'anshan, Zibo, Linyi, Luoyang, Nanyang, Aba Tibetan and Qiang Autonomous Prefecture, Lijiang, Zhangye City; County-level cities: Xiaoyi City, Dashiqiao City, Longkou City, Laizhou City; Counties/autonomous counties: Anyang County, Yunyang County, Shangri-La County. |

**Table A2.** GDE in China's 112 resource-based cities from 2010 to 2019.

| DMUs | GDE | | | | | | | | | |
|---|---|---|---|---|---|---|---|---|---|---|
| | 2010 | 2011 | 2012 | 2013 | 2014 | 2015 | 2016 | 2017 | 2018 | 2019 |
| DMU1 | 0.7503 | 0.7927 | 0.7441 | 0.7707 | 0.7648 | 0.8093 | 0.7391 | 0.7910 | 0.8107 | 1.0558 |
| DMU2 | 0.6897 | 0.7088 | 0.7125 | 0.6780 | 0.6859 | 0.7519 | 0.6798 | 0.7220 | 0.7052 | 0.8239 |
| DMU3 | 0.6524 | 0.6504 | 0.6399 | 0.6434 | 0.6753 | 0.7270 | 0.6345 | 0.7281 | 0.7168 | 1.0427 |
| DMU4 | 0.6117 | 0.6158 | 0.5935 | 0.6245 | 0.6262 | 0.6730 | 0.6479 | 0.7171 | 0.6996 | 0.7586 |
| DMU5 | 0.6510 | 0.6939 | 0.6793 | 0.7081 | 0.7310 | 0.7721 | 0.6476 | 0.6918 | 0.7024 | 0.7617 |
| DMU6 | 0.6103 | 0.5568 | 0.5333 | 0.5307 | 0.5395 | 0.5848 | 0.6092 | 0.7464 | 0.7611 | 1.0713 |
| DMU7 | 0.5868 | 0.5414 | 0.6044 | 0.5841 | 0.5657 | 0.5917 | 0.5834 | 0.6780 | 0.7074 | 1.1228 |
| DMU8 | 0.6757 | 0.6770 | 0.6444 | 0.6467 | 0.6538 | 0.6336 | 0.5730 | 0.7911 | 0.7633 | 0.7907 |
| DMU9 | 0.7711 | 0.7477 | 0.6895 | 0.7010 | 0.6982 | 0.6636 | 0.5896 | 0.8291 | 0.8323 | 0.9375 |
| DMU10 | 0.7474 | 0.7200 | 0.7172 | 0.7172 | 0.7347 | 0.7256 | 0.7851 | 1.2265 | 1.2124 | 0.8879 |

**Table A2.** *Cont.*

| DMUs | GDE | | | | | | | | | |
|------|------|------|------|------|------|------|------|------|------|------|
| | **2010** | **2011** | **2012** | **2013** | **2014** | **2015** | **2016** | **2017** | **2018** | **2019** |
| DMU11 | 0.7327 | 0.7087 | 0.6314 | 0.6531 | 0.6498 | 0.6518 | 0.6184 | 0.9369 | 0.9018 | 1.1167 |
| DMU12 | 0.7116 | 0.7277 | 0.6593 | 0.6781 | 0.6794 | 0.6473 | 0.6078 | 0.7002 | 0.7055 | 0.7659 |
| DMU13 | 0.6397 | 0.6464 | 0.5988 | 0.6331 | 0.6284 | 0.5871 | 0.4917 | 0.7744 | 0.7247 | 1.3994 |
| DMU14 | 0.9123 | 0.8001 | 0.6762 | 0.7075 | 0.7133 | 0.7450 | 0.6027 | 0.6972 | 0.7223 | 0.7713 |
| DMU15 | 1.1112 | 1.1363 | 1.2315 | 1.0132 | 1.0933 | 0.7150 | 0.6434 | 1.2155 | 1.0411 | 1.6700 |
| DMU16 | 0.8546 | 0.7705 | 0.7900 | 0.7470 | 0.7914 | 0.8781 | 0.9385 | 0.7389 | 0.7822 | 0.8718 |
| DMU17 | 0.6112 | 0.5834 | 0.6608 | 0.6177 | 0.6243 | 0.6421 | 1.0321 | 0.6552 | 0.7126 | 0.7741 |
| DMU18 | 0.7242 | 0.6479 | 0.6695 | 0.6835 | 0.7078 | 0.7456 | 0.7153 | 0.7417 | 0.7342 | 0.8071 |
| DMU19 | 1.3796 | 1.3048 | 1.2613 | 1.3487 | 1.4886 | 1.4335 | 1.4305 | 1.2982 | 1.2812 | 1.3056 |
| DMU20 | 0.7671 | 0.8609 | 0.7444 | 0.8057 | 1.1083 | 0.8414 | 0.8659 | 0.8364 | 0.6975 | 1.0827 |
| DMU21 | 0.8049 | 0.6821 | 0.6643 | 0.6517 | 0.6143 | 0.6833 | 0.7089 | 0.8158 | 0.8297 | 0.8311 |
| DMU22 | 0.6840 | 0.6425 | 0.6491 | 0.6461 | 0.6149 | 0.7245 | 1.0878 | 1.0445 | 1.0717 | 0.8170 |
| DMU23 | 0.6752 | 0.6072 | 0.5930 | 0.5838 | 0.5449 | 0.6587 | 0.6567 | 0.7396 | 0.7533 | 0.7181 |
| DMU24 | 0.5782 | 0.5086 | 0.5213 | 0.5404 | 0.5552 | 0.6170 | 0.6986 | 0.6581 | 0.7459 | 0.8629 |
| DMU25 | 0.7933 | 0.7317 | 0.6464 | 0.6533 | 0.6651 | 0.7397 | 0.6749 | 0.9121 | 0.7955 | 0.9048 |
| DMU26 | 0.6309 | 0.6306 | 0.5626 | 0.6274 | 0.6073 | 0.7479 | 1.0032 | 0.8048 | 0.8928 | 1.1033 |
| DMU27 | 0.7649 | 0.8428 | 0.8669 | 0.8132 | 0.7737 | 0.8259 | 0.8166 | 1.0589 | 0.7855 | 0.7351 |
| DMU28 | 0.5982 | 0.6812 | 0.6429 | 0.6971 | 0.6962 | 0.6356 | 0.6743 | 0.7776 | 0.6887 | 0.6690 |
| DMU29 | 0.6868 | 0.7529 | 0.7356 | 0.7452 | 0.7393 | 0.7876 | 0.7362 | 0.8871 | 0.8254 | 0.7798 |
| DMU30 | 1.5239 | 1.4868 | 1.4824 | 1.5619 | 1.2827 | 1.3780 | 1.2875 | 1.3029 | 1.4409 | 0.7727 |
| DMU31 | 0.7641 | 0.6667 | 0.6808 | 0.6947 | 0.7534 | 0.7151 | 0.7988 | 0.7504 | 0.7374 | 0.8092 |
| DMU32 | 0.6009 | 0.6059 | 0.6139 | 0.6235 | 0.6061 | 0.5973 | 0.5944 | 0.6115 | 0.6158 | 0.6298 |
| DMU33 | 0.6732 | 0.6805 | 0.6686 | 0.6902 | 0.7042 | 0.7744 | 1.1196 | 1.1522 | 0.7561 | 0.6909 |
| DMU34 | 1.1759 | 1.2499 | 1.2206 | 1.2297 | 1.4564 | 1.4926 | 1.2756 | 1.3074 | 1.1439 | 0.9919 |
| DMU35 | 0.6818 | 0.5779 | 0.5504 | 0.5735 | 0.5904 | 0.6392 | 0.6984 | 0.6853 | 0.6955 | 0.8115 |
| DMU36 | 0.6827 | 0.6787 | 0.6805 | 0.5998 | 0.5111 | 0.5634 | 0.8044 | 0.7116 | 0.7380 | 0.7581 |
| DMU37 | 0.7100 | 0.8224 | 0.7794 | 0.8986 | 1.0126 | 1.0375 | 0.6872 | 1.2737 | 1.0005 | 1.3816 |
| DMU38 | 0.8476 | 0.7608 | 0.6955 | 0.8560 | 1.0332 | 0.8855 | 0.8783 | 1.0633 | 1.0167 | 1.3086 |
| DMU39 | 0.7628 | 0.7595 | 0.7462 | 0.7303 | 0.7604 | 0.8581 | 0.7337 | 0.9831 | 1.0506 | 1.1350 |
| DMU40 | 0.7454 | 0.8304 | 0.8357 | 0.8317 | 0.8246 | 1.0075 | 0.7361 | 0.7989 | 0.7495 | 0.7644 |
| DMU41 | 0.7386 | 0.7267 | 0.6947 | 0.7370 | 0.7338 | 0.8106 | 0.7225 | 0.7375 | 0.7080 | 0.7570 |
| DMU42 | 0.5996 | 0.5539 | 0.5717 | 0.5359 | 0.5272 | 0.5535 | 0.5517 | 0.5855 | 0.5955 | 0.6633 |
| DMU43 | 0.7106 | 0.7019 | 0.7044 | 0.6428 | 0.6502 | 0.7122 | 0.6160 | 0.7143 | 0.6706 | 0.6921 |
| DMU44 | 0.6231 | 0.5914 | 0.5970 | 0.6161 | 0.6111 | 0.5910 | 0.5721 | 0.6929 | 0.6813 | 1.0017 |
| DMU45 | 0.6473 | 0.6385 | 0.6179 | 0.5739 | 0.5877 | 0.6996 | 0.6388 | 0.7256 | 0.6959 | 0.5773 |
| DMU46 | 0.6731 | 0.7908 | 0.7641 | 0.7705 | 0.8119 | 0.8295 | 0.7320 | 0.8398 | 0.6846 | 0.8004 |
| DMU47 | 0.7632 | 0.7066 | 0.7161 | 0.7353 | 0.7400 | 0.7718 | 0.6926 | 0.8317 | 0.8303 | 1.0029 |
| DMU48 | 0.8424 | 0.8405 | 0.8252 | 0.9173 | 0.9098 | 0.8651 | 0.7865 | 0.7649 | 0.7300 | 0.8364 |
| DMU49 | 0.6700 | 0.7060 | 0.7156 | 0.6860 | 0.6768 | 0.7789 | 0.7209 | 0.7585 | 0.7079 | 0.7553 |
| DMU50 | 0.6801 | 0.7037 | 0.6151 | 0.6911 | 0.8029 | 0.8267 | 0.7691 | 0.8263 | 0.7397 | 0.7790 |
| DMU51 | 0.6560 | 0.6799 | 0.6629 | 0.7108 | 0.7295 | 0.8151 | 0.7057 | 1.0576 | 1.0853 | 1.0894 |
| DMU52 | 0.6721 | 0.7217 | 0.7077 | 0.7602 | 0.7841 | 0.8044 | 0.7813 | 0.8492 | 0.8371 | 1.0186 |
| DMU53 | 0.7705 | 0.7705 | 0.7952 | 0.8172 | 0.8644 | 0.9007 | 0.7371 | 0.8549 | 0.8202 | 0.8628 |
| DMU54 | 0.6518 | 0.6518 | 0.6452 | 0.6883 | 0.6943 | 0.6610 | 0.6302 | 0.6254 | 0.6445 | 0.8874 |
| DMU55 | 0.6130 | 0.6130 | 0.6062 | 0.6154 | 0.6358 | 0.6513 | 0.6220 | 0.7084 | 0.7155 | 0.6960 |
| DMU56 | 0.7125 | 0.7125 | 0.6915 | 0.6467 | 0.6686 | 0.7049 | 0.6737 | 0.7281 | 0.7281 | 0.6930 |
| DMU57 | 0.7551 | 0.7551 | 0.7240 | 0.7691 | 0.7143 | 0.7083 | 0.6423 | 0.6837 | 0.6659 | 0.7315 |
| DMU58 | 0.7278 | 0.7278 | 0.7163 | 0.7564 | 0.7679 | 0.7633 | 0.6804 | 0.7101 | 0.6988 | 0.7187 |
| DMU59 | 0.6802 | 0.6802 | 0.6683 | 0.6785 | 0.6659 | 0.7271 | 0.6664 | 0.7398 | 0.7235 | 0.7157 |
| DMU60 | 0.7179 | 0.7179 | 0.6832 | 0.7073 | 0.7103 | 0.7433 | 0.6838 | 0.8086 | 0.8646 | 0.6860 |
| DMU61 | 1.0025 | 1.0025 | 0.8504 | 0.9374 | 1.0085 | 1.0505 | 0.7856 | 1.0592 | 1.1188 | 0.8027 |
| DMU62 | 0.7789 | 0.7789 | 0.7335 | 0.7472 | 0.7539 | 0.8164 | 0.7332 | 0.7572 | 0.8526 | 0.7814 |
| DMU63 | 0.7997 | 0.7997 | 0.7728 | 0.9209 | 0.9769 | 1.0033 | 0.8146 | 1.0310 | 1.0180 | 0.7693 |
| DMU64 | 0.7666 | 0.7666 | 0.7269 | 0.7290 | 0.7156 | 0.8194 | 0.6842 | 0.7292 | 0.7289 | 0.7718 |
| DMU65 | 0.6979 | 0.6979 | 0.7123 | 0.7210 | 0.7257 | 0.7674 | 0.7315 | 0.8763 | 0.8332 | 0.9062 |
| DMU66 | 0.6858 | 0.6858 | 0.6164 | 0.6248 | 0.6275 | 0.6698 | 0.7261 | 0.7023 | 0.6976 | 0.9326 |
| DMU67 | 0.5981 | 0.5981 | 0.5913 | 0.6065 | 0.5992 | 0.6704 | 0.6420 | 0.6787 | 0.6560 | 1.0267 |

**Table A2.** *Cont.*

| DMUs | GDE | | | | | | | | | |
|---|---|---|---|---|---|---|---|---|---|---|
| | **2010** | **2011** | **2012** | **2013** | **2014** | **2015** | **2016** | **2017** | **2018** | **2019** |
| DMU68 | 0.6940 | 0.6940 | 0.6597 | 0.6841 | 0.6735 | 0.7201 | 0.6716 | 0.7545 | 0.7621 | 1.1090 |
| DMU69 | 0.7186 | 0.7186 | 0.6548 | 0.6971 | 0.7116 | 0.7595 | 0.6946 | 0.7716 | 0.7264 | 0.9355 |
| DMU70 | 0.7073 | 0.7073 | 0.6789 | 0.6934 | 0.7049 | 0.7110 | 0.6686 | 0.9281 | 1.0212 | 0.8764 |
| DMU71 | 0.7778 | 0.7778 | 0.7254 | 0.7822 | 0.7651 | 0.8772 | 0.8939 | 1.0396 | 0.9622 | 1.0360 |
| DMU72 | 0.5700 | 0.5700 | 0.5991 | 0.6093 | 0.6104 | 0.6401 | 0.6361 | 0.6651 | 0.6666 | 0.7447 |
| DMU73 | 0.6409 | 0.6409 | 0.5692 | 0.6410 | 0.6535 | 0.6601 | 0.7026 | 0.6777 | 0.6896 | 0.8209 |
| DMU74 | 0.7453 | 0.7453 | 0.7237 | 0.7424 | 0.7519 | 0.8043 | 0.7111 | 0.7953 | 0.7877 | 1.0115 |
| DMU75 | 0.7953 | 0.7953 | 0.9938 | 0.7814 | 0.8197 | 0.9262 | 0.8857 | 0.8071 | 0.8209 | 0.9731 |
| DMU76 | 0.6961 | 0.6961 | 0.7123 | 0.7504 | 0.7854 | 0.8269 | 0.7658 | 0.8240 | 0.7744 | 0.7976 |
| DMU77 | 0.7196 | 0.7196 | 0.7654 | 0.6813 | 0.6820 | 0.7073 | 0.6913 | 0.7489 | 0.7081 | 0.7186 |
| DMU78 | 0.6302 | 0.6302 | 0.6256 | 0.6186 | 0.6195 | 0.7064 | 0.6612 | 0.6730 | 0.6647 | 0.6088 |
| DMU79 | 0.6744 | 0.6744 | 0.6568 | 0.7200 | 0.7310 | 1.0158 | 0.6720 | 0.6897 | 0.6837 | 0.7268 |
| DMU80 | 0.5726 | 0.5726 | 0.5441 | 0.5916 | 0.6567 | 0.7059 | 0.6978 | 0.7029 | 0.6286 | 0.6931 |
| DMU81 | 0.6148 | 0.6148 | 0.6536 | 0.6802 | 0.6811 | 0.6954 | 0.7092 | 0.7040 | 0.7107 | 0.9004 |
| DMU82 | 0.5723 | 0.5723 | 1.0003 | 0.6704 | 1.1634 | 1.1326 | 1.1915 | 0.6569 | 0.6811 | 0.6690 |
| DMU83 | 0.9334 | 0.9334 | 0.7752 | 0.8410 | 1.0459 | 1.0964 | 1.1289 | 1.1159 | 1.1604 | 1.1604 |
| DMU84 | 0.5336 | 0.5336 | 0.5652 | 0.5584 | 0.5064 | 0.5677 | 0.5863 | 0.6113 | 0.6108 | 0.6740 |
| DMU85 | 0.7019 | 0.7019 | 0.7154 | 0.7319 | 0.7267 | 0.7661 | 0.6430 | 0.7029 | 0.6483 | 0.7151 |
| DMU86 | 0.6055 | 0.6055 | 0.6935 | 0.7098 | 0.7250 | 0.7810 | 0.8014 | 0.9311 | 0.7565 | 1.0199 |
| DMU87 | 0.9455 | 0.9455 | 0.9530 | 1.0299 | 1.1213 | 1.1287 | 0.8229 | 1.0336 | 1.3314 | 1.1417 |
| DMU88 | 1.0458 | 1.0458 | 0.7723 | 0.7363 | 0.8070 | 0.8542 | 0.6517 | 0.8942 | 0.8207 | 0.8668 |
| DMU89 | 0.7488 | 0.7488 | 0.7755 | 0.9301 | 0.7427 | 0.8025 | 0.6936 | 0.7810 | 0.7367 | 1.3063 |
| DMU90 | 0.7346 | 0.7346 | 0.8530 | 0.8337 | 0.7953 | 0.9144 | 0.7718 | 0.8294 | 0.7767 | 0.9320 |
| DMU91 | 0.6124 | 0.6124 | 0.5501 | 0.5694 | 0.5729 | 0.6640 | 0.6153 | 0.6361 | 0.6716 | 1.0101 |
| DMU92 | 0.5620 | 0.5620 | 0.5390 | 0.6406 | 0.6527 | 0.7142 | 0.6980 | 0.6913 | 0.6378 | 0.6170 |
| DMU93 | 0.6666 | 0.6666 | 0.6834 | 0.7358 | 0.7496 | 0.7205 | 0.6132 | 0.6616 | 0.6274 | 0.7955 |
| DMU94 | 0.6085 | 0.6085 | 0.6610 | 0.7115 | 0.7012 | 0.6829 | 0.5770 | 0.7162 | 0.6748 | 1.0786 |
| DMU95 | 0.7676 | 0.7676 | 0.6672 | 0.7355 | 0.7641 | 0.8584 | 0.7232 | 0.7513 | 0.6481 | 0.8071 |
| DMU96 | 0.6786 | 0.6786 | 0.5664 | 0.6483 | 0.7539 | 0.8057 | 0.7165 | 0.6977 | 0.6324 | 0.8274 |
| DMU97 | 0.6352 | 0.6352 | 0.7008 | 0.6916 | 0.6923 | 0.6701 | 0.6506 | 1.1442 | 1.0336 | 1.0821 |
| DMU98 | 0.7021 | 0.7021 | 0.6437 | 0.7017 | 0.6721 | 0.7004 | 0.6172 | 0.6278 | 0.5640 | 0.6833 |
| DMU99 | 0.7167 | 0.7167 | 0.7592 | 0.8069 | 0.7764 | 0.8095 | 0.7177 | 0.8470 | 1.0080 | 0.8768 |
| DMU100 | 1.0416 | 1.0416 | 1.0708 | 1.0513 | 1.0144 | 1.1169 | 0.9413 | 0.9019 | 0.7786 | 1.0121 |
| DMU101 | 1.0551 | 1.0551 | 1.0435 | 0.7585 | 0.7226 | 0.7474 | 0.6564 | 0.6824 | 0.6541 | 0.6766 |
| DMU102 | 1.1312 | 1.1312 | 1.0456 | 1.1557 | 1.1112 | 1.0074 | 0.7734 | 0.8431 | 0.8729 | 1.0609 |
| DMU103 | 1.2217 | 1.2217 | 1.0829 | 1.1312 | 1.1706 | 1.1668 | 1.1549 | 1.0780 | 1.0308 | 1.7653 |
| DMU104 | 0.5598 | 0.5598 | 0.5304 | 0.5307 | 0.4753 | 0.4554 | 0.5359 | 0.5670 | 0.5571 | 0.6158 |
| DMU105 | 0.5072 | 0.5072 | 0.5071 | 0.7759 | 0.4027 | 0.4390 | 0.4883 | 0.9381 | 0.7318 | 0.7854 |
| DMU106 | 0.8887 | 0.8887 | 0.9920 | 0.5717 | 0.6225 | 0.6988 | 0.7298 | 0.8159 | 0.9024 | 0.8873 |
| DMU107 | 0.6467 | 0.6467 | 0.6488 | 0.7819 | 0.6363 | 0.6752 | 0.6692 | 0.7281 | 0.7176 | 0.7355 |
| DMU108 | 0.5933 | 0.5933 | 0.5139 | 0.5658 | 0.6064 | 0.5947 | 0.5459 | 0.6220 | 0.6479 | 0.8326 |
| DMU109 | 1.2358 | 1.2358 | 1.2133 | 1.1426 | 1.2290 | 1.3603 | 1.1359 | 1.1042 | 1.1733 | 1.1216 |
| DMU110 | 1.0563 | 1.0563 | 0.7216 | 0.7292 | 0.7175 | 0.8799 | 0.6684 | 0.9119 | 0.7439 | 0.8542 |
| DMU111 | 0.5741 | 0.5741 | 0.5445 | 0.5369 | 0.5459 | 0.5944 | 0.5792 | 0.6303 | 0.5948 | 0.6914 |
| DMU112 | 1.0231 | 1.0231 | 0.7370 | 0.6621 | 0.6713 | 0.6775 | 0.7170 | 0.7838 | 0.8149 | 0.7757 |

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
