# Peer review of "The Effect of Industrial Structure Upgrading and Human Capital Structure Upgrading on Green Development Efficiency—Based on China’s Resource-Based Cities"

_sustainability, doi:10.3390/su15054673_

Round 1
Reviewer 1 Report
I reviewed the given manuscript on “The direct effect and interaction effect of Industrial Structure Upgrading and Human Capital Structure Upgrading on Green Development Efficiency of China’s Resource-based Cities”, while my major comments are given. These comments are directly linked with the given study and will improve the quality of the paper.
Major comments are given below,
My first concern is to add the highlights of this study. I think without highlights study is incomplete. Similarly, at this stage the title of paper is not appropriate please re-think and change the title.
Abstract:
· The Abstract has minor significant information for a reader; therefore, please follow the recent studies to write a catchy abstract.
· Start your abstract from the problem and add three to four sentences i.e., problem and initiative for it. Please try to write your abstract via some numerical logics the green development situation.
· It is highly recommended to change your abstract in present era.
Introduction
· Before to start up new paragraph (Page 2; line 48); the addition of green development in introduction section is not clear. It is recommended that author must clear why there is need of such initiative? Provide at least 8 to 10 lines.
· The first paragraph of the study has no significant information, try to link with crux why you have focused on concept of environment, sustainability, and relevant perspectives. Therefore, it is necessary to discuss about the energy and environmental situation under the perspectives of concerned issues. Before, move forward to problem author must try to elaborate how this connects with environment and they are interlink with each other and can affect the environmental quality under the theme of sustainability and energy transition.
· However, it will be excellent to start up your open paragraph with global issue and it does interlink with development and energy. Furthermore, you have to clear what is the best solution such as green development and then move to China and President Initiatives.
· Please avoid the report references or sentences, it will be better to cite from any paper.
· Moreover, the research gap is unclear; try to improve it according to journal requirements.
· Please start your intro and you should introduce all abbreviation in the start of text.
· It is strongly recommended to introduce the clear association of energy, industrial structure up-gradation, sustainable/green development and environment. You can see the recent studies,
doi/10.1007/s11356-020-10485-w; https://doi.org/10.1016/j.apenergy.2022.118522; https://doi.org/10.1016/j.rser.2020.110244; https://doi.org/10.1016/j.ecolind.2021.107638,
https://doi.org/10.1016/j.landusepol.2022.106224.
Literature Review:
· Reader may not feel friendly in reading; therefore it is necessary to include a separate section on literature review with different headings.
· Please add some recent studies and try to add literature section on spread effects of environmental issues and socio-economic response towards environmental sustainability.
· In addition (literature review), the authors should introduced a proper channel of studies that must deliver a clear message to the reviewer. Therefore, it is necessary to relate your manuscript to the ex-studies on this theme, gender activities, energy, and environment situation, besides the simple one. Try to include studies from 2019-2022 not only for specified economy but also other regions, like Oman, Qatar, India, Bangladesh and Pakistan Besides, follow the given studies to deliver a clear message to the reader. https://doi.org/10.3390/su132011430; https://doi.org/10.1016/j.energy.2022.124377; doi/10.1007/s10668-021-02013-8; https://doi.org/10.1016/j.renene.2022.07.008.
Data and Method
· Please concise this section in a very clear way.
· However. Authors have made a great effort and try to clear how many studies have been done on such issues and their solutions. But there is need to cite recent studies.
· Add a graphical presentation for estimation strategy.
· R2 is minor please discuss the logics.
Results
· Please cite some recent studies and their comments regarding your outcomes. The discussion section is weak; therefore, authors must focus on this section.
· If possible please add one paragraph of comparison your outcomes with previous one.
· Add graphical presentation of your outcomes in one PPT graph.
Conclusion
· In the conclusion section there is need to re-concise conclusion section. I think these are your findings; therefore, please re-write or compile into three sentences.
· Policy recommendations are not clear at this stage of the manuscript. I think the authors must add a one line of variable relation then suggest the policy in conclusion section.
· Also, add some exciting limitations regarding political factors for the future studies. At this stage there are very limited. Would you like to share your opinions concerning governance and its association with green development, industrial structure up-gradation?
Reviewer 2 Report
I read the manuscript. The subject of the manuscript is very interesting. But the authors have not focused on the technical discussion of the subject for evaluation. This position can be easily seen in Model 5. Using undesired input in output is an outdated attitude in data envelopment analysis.
Also, the type of second input is Integer that the modeling should be done in such a way that the benchmark is Integer, but there is no such thing in the manuscript.
Considering the lack of innovation in modeling and the problems stated above, my suggestion is to reject the manuscript.
Reviewer 3 Report
The English must be improved by native speakers.
Abstract: please add some quantitative results in the abstract section.
Line 32-33, please remove this sentence and add one on the research significance of your paper.
Please remove line 41-46, but you can add more information on the sustainable development goals proposed by the United Nations. The description of sustainable development goals can be more attractive for international readers.
Line 48-50, such a description is too broad and vague. Could you please explicitly define green development?
After line 64 in the introduction, the information is not attractive, and authors have not well defined the research gap of existing studies. Therefore, authors have to rewrite the introduction, and presenting existing studies on the industrial structure (improvement) and human capital structure is a good start.
Section 2 is not well written, and why did you propose such hypothesis?
Subsection 3.1.1 what are the 126 resource-based cities? Please present a map to locate them.
Table 1, are they the variable defined by authors themselves? Any sources or references? Could you add some more social indicators, such as the wellbeing, safe, and health?
Authors have to add some discussion on the sustainable development, industrial structure, human capital and future urban governance.
Section 6 should be the section of Limitation and future studies, rather than a discussion section.
Authors have purely cited Chinese papers which are highly unacceptable. Please update. The following references are meaningful:
The linkage between sustainable development goals 9 and 11: Examining the association between sustainable urbanization and intellectual property rights protection. Advanced Sustainable Systems, 6(3), 2100283.
Promoting and implementing urban sustainability in China: An integration of sustainable initiatives at different urban scales. Habitat International, 82, 83-93.
Industrial economics: issues and perspectives. Bloomsbury Publishing.
What drives the implementation of Industry 4.0? The role of opportunities and challenges in the context of sustainability. Sustainability, 10(1), 247.
Influences of the industry 4.0 revolution on the human capital development and consumer behavior: A systematic review. Sustainability, 12(10), 4035
Economic growth, human capital and structural change: A dynamic panel data analysis. Research policy, 45(8), 1636-1648.
Reviewer 4 Report
1. Title can be shortened. Too many repetitive words.
2. For clarity, structure the literature review into its different aspects with proper headings. Create a separate section for the literature review. This will help readers understand the existing research conducted on this topic.
3. Why are ISU and HCSU the most crucial aspects of GDE? What are the other aspects?
4. The recommendations provided are rather high-level. Execution is a critical part of policymaking. Perhaps some techniques used by the government can be emphasized to ensure the provided suggestions have a realistic outcome.
5. How do you verify the directions recommended for cities to take? What are the different tools that can be used to evaluate the results? Are there any simulation tools used for validation in this study?
6. How would the results of this study apply to other countries and industries? What are the different aspects to consider?
7. Provide more details on the potential research directions of this study.
Round 2
Reviewer 1 Report
Thank you very much for giving me opportunity to review again modified manuscript. Authors have significant changes and they satisfied what was asking; therefore, i want to accept the given updated manuscript.
Reviewer 3 Report
well done
Reviewer 4 Report
1. There are still spelling errors, and mistakes spotted in the manuscript. Please carefully revise. E.g., Line 28 “different resouece-based cities”, Line 671 (HUSU and ISU simultaneously should be promoted simultaneously.) Also be careful of wordings. E.g., Line 74 (The research objects on green development …)
2. Do consider removing the abbreviations table or removing full names from the manuscript for those abbreviated in the table. Include the highlights in the introduction. Reduce repetition in the paper.
3. Try not to repeat points from the literature review in the introduction. Also, in Line 178, what is the subsection referred to?
4. A GDE value of 0.8, which is considered relatively low, is stated in the abstract. This value alone doesn’t give the reader a good grasp of how low it is. Please provide a comparison.